# Formation of Highly Conductive Interfaces in Crystalline Ionic Liquid-Gated Unipolar MoTe_2_/h-BN Field-Effect Transistor

**DOI:** 10.3390/nano13182559

**Published:** 2023-09-15

**Authors:** Kamoladdin Saidov, Jamoliddin Razzokov, Odilkhuja Parpiev, Nur Sena Yüzbasi, Natalia Kovalska, Gurdial Blugan, Olim Ruzimuradov

**Affiliations:** 1Department of Electronics and Radio Engineering, Tashkent University of Information Technologies, Tashkent 100200, Uzbekistan; 2Department of Information Technologies, Tashkent International University of Education, Tashkent 100207, Uzbekistan; 3Department of Electrical and Computer Engineering, Ajou University in Tashkent, Tashkent 100204, Uzbekistan; 4R&D Center, New Uzbekistan University, Tashkent 100007, Uzbekistan; jrazzokov@gmail.com; 5School of Engineering, Central Asian University, Tashkent 111221, Uzbekistan; 6Institute of Fundamental and Applied Research, National Research University TIIAME, Tashkent 100000, Uzbekistan; 7Material Sciences Institute, Academy of Sciences of the Republic of Uzbekistan, Tashkent 100084, Uzbekistan; o.parpiev@imssolar.uz; 8Laboratory for High Performance Ceramics, Empa, Swiss Federal Laboratories for Materials Science and Technology, 8600 Dübendorf, Switzerland; sena.yuezbasi@empa.ch (N.S.Y.); natalia.kovalska@empa.ch (N.K.); 9Department of Natural and Mathematic Sciences, Turin Polytechnic University in Tashkent, Tashkent 100095, Uzbekistan

**Keywords:** ionic-liquid gating, MoTe2/h-BN field-effect transistor, transconductance, charge transfer

## Abstract

2H MoTe_2_ (molybdenum ditelluride) has generated significant interest because of its superconducting, nonvolatile memory, and semiconducting of new materials, and it has a large range of electrical properties. The combination of transition metal dichalcogenides (TMDCs) and two dimensional (2D) materials like hexagonal boron nitride (h-BN) in lateral heterostructures offers a unique platform for designing and engineering novel electronic devices. We report the fabrication of highly conductive interfaces in crystalline ionic liquid-gated (ILG) field-effect transistors (FETs) consisting of a few layers of MoTe_2_/h-BN heterojunctions. In our initial exploration of tellurium-based semiconducting TMDs, we directed our attention to MoTe_2_ crystals with thicknesses exceeding 12 nm. Our primary focus centered on investigating the transport characteristics and quantitatively assessing the surface interface heterostructure. Our transconductance (*g_m_*) measurements indicate that the very efficient carrier modulation with an ILG FET is two times larger than standard back gating, and it demonstrates unipolarity of the device. The ILG FET exhibited highly unipolar p-type behavior with a high on/off ratio, and it significantly increased the mobility in MoTe_2_/h-BN heterochannels, achieving improvement as one of the highest recorded mobility increments. Specifically, we observed hole and electron mobility values ranging from 345 cm^2^ V^−1^ s^−1^ to 285 cm^2^ V^−1^ s^−1^ at 80 K. We predict that our ability to observe the intrinsic, heterointerface conduction in the channels was due to a drastic reduction of the Schottky barriers, and electrostatic gating is suggested as a method for controlling the phase transitions in the few layers of TMDC FETs. Moreover, the simultaneous structural phase transitions throughout the sample, achieved through electrostatic doping control, presents new opportunities for developing phase change devices using atomically thin membranes.

## 1. Introduction

MoTe_2_ (molybdenum ditelluride) is a two-dimensional material that has gained significant interest in the field of electronics because of its unique electronic properties [1,2]. Another noteworthy two-dimensional material is h-BN, which is composed of boron and nitrogen atoms arranged in a hexagonal lattice structure, similar to graphene. However, unlike graphene, h-BN is an insulator with a wide bandgap, making it an excellent electrical insulator [3]. Morever, h-BN possesses several desirable properties, such as high thermal conductivity, excellent mechanical strength, and chemical stability. These characteristics make h-BN an ideal material for various applications, including protective coatings, thermal management in electronic devices, and as a substrate for growing other 2D materials [4]. In the realm of two-dimensional materials, lateral heterostructures play a crucial role. These structures involve combining different layers of 2D materials in a lateral arrangement, stacking them together to create a heterostructure. By stacking alternating layers of TMDCs and h-BN, a lateral heterostructure comprising TMDCs and h-BN can be created [5]. This integration enables the exploration of new properties and potential applications arising from the synergistic effects among the constituent materials. The insulating nature of h-BN serves as a barrier layer, effectively preventing electrical leakage between the TMDC layers. This allows for the development of high-performance electronic devices with enhanced functionality and reduced power consumption. One way to modulate the electrical behavior of MoTe_2_ is through FETs using ILG. ILG is a technique that utilizes an ionic liquid as the gate dielectric, replacing the use of traditional solid-state dielectrics [6,7,8,9]. The choice of IL is crucial, as it should have a high ionic conductivity and be compatible with the material being gated. In the case of MoTe_2_/h-BN FETs, unipolar ILG refers to the use of an ionic liquid that predominantly contains either positive or negative ions. However, achieving unipolar n-type behavior in devices is quite challenging because of a combination of factors, including Fermi-level pinning and the limited availability of low-work function contacts [10]. The formation of highly conductive interfaces in MoTe_2_/h-BN FETs with unipolar ILG is achieved by applying a voltage bias between the source and drain electrodes, while the ionic liquid acts as the gate dielectric. When a positive voltage bias is applied, positive ions from the ionic liquid accumulate near the MoTe_2_’s surface, resulting in the formation of a conductive hole accumulation layer. Conversely, when a negative voltage bias is applied, negative ions accumulate, creating an electron accumulation layer [8,11].

The accumulation of either holes or electrons at the MoTe_2_ surface modifies its conductivity [12]. By adjusting the voltage bias, it is possible to tune the carrier density in the MoTe_2_ channel, effectively switching the device between an on state (i.e., high conductivity) and an off state (i.e., low conductivity). This ability to control the conductivity of MoTe_2_/h-BN through ILG makes it a promising technology for flexible and low-power electronic applications. The field of ILGs is still an area of active research, and ongoing developments continue to improve the understanding and performance of devices based on this concept. Nanomaterials are contemporary materials in demand for the fabrication of multifunctional, lightweight, and portable modern devices [13,14,15,16,17]. The application range of these kinds of devices is quite broad, including biosensing, toxicity detection, food analysis, electronic skins, flexible displays, highly conductive materials, and solar cells [18,19,20]. The core component of these devices consist of FETs [21,22]. An FET maintains three terminals: source, gate, and drain, and this enables for the control of the flow of the charge in semiconductors [23]. MoTe_2_-based FETs have shown beneficial electrical properties for employment in the engineering of semiconductor materials [24,25].

The major desired feature of semiconductors is to maintain a low bandgap energy. This ranges from 0.88 to 1.1 eV in MoTe_2_ with respect to its lattice arrangements and thickness [26,27]. Therefore, MoTe_2_ is a better candidate for use with h-BN in optoelectronic devices that operate in the near-infrared wavelength range in comparison to other materials [28,29]. Thus, MoTe_2_ can be exploited in the manufacturing of FETs because of the small bandgap, which enables the large tunneling current and high on–off ratio. The performance of MoTe_2_-based FETs can further be enhanced by using ILs as a gate dielectric material [30,31,32].

ILs are a new kind of electrolyte that differ from aqueous, organic, gel, and polymer electrolytes [33]. The thermal and chemical stability of ILs is quite high. Moreover, ILs remain in a fluid state over a broad range of temperatures [34]. Most importantly, wide electrochemical windows blockade occurring redox reactions even at high potential differences [35]. ILs, such as DEME-TFSI (diethylmethyl(2-methoxyethyl)ammonium bis(trifluoromethylsulfonyl)imide), DEME-BF_4_ (diethylmethyl-(2-methoxyethyl)ammonium tetrafluoroborate), BMIM-PF_6_ (1-butyl-3-methylimidazolium hexafluorophosphate), BMIM-TFSI (1-butyl-3-methylimidazolium bis(trifluoromethylsulfonyl)imide), and BMPPD-TFSI (bis(trifluoromethylsulfonyl)imide), have been used as gate dielectric material in combination with MoS_2_ (molybdenum disulfide), MoSe_2_ (molybdenum(IV) selenide), and MoTe_2_, and the MoS_2_/h-BN heterostructure was observed, according to the literature [36,37,38,39,40,41,42]. For example, the measured electron and hole sheet carrier densities showed an n-type conductivity in the monolayer MoTe_2_-DEME-TFSI system [37]. In contrast, an EDLT structure and DEME-TFSI employed as a gate dielectric on 20–100 nm thick MoTe_2_ resulted in the obtained FETs exhibited an ambipolar character [40]. Thus, the nature of the charge carrier ability of MoTe_2_/h-BN might be controlled by changing its thickness and introducing various kind of ILs. Hence, the production of FETs with the desired parameters can be achieved by tuning the MoTe_2_/h-BN’s thickness and the use of ILs. Researchers have successfully obtained results that demonstrate the significance of unipolar p/n-type FETs as fundamental units for use in various functional electronics and optoelectronics applications. Previous studies focused on investigating the ideal chemical dopants for MoTe_2_ [9,43,44], while surface charge transfer methods have been utilized for the electron doping of a few or multiple layers of MoTe_2_ [45]. In our case, we observed the maximum value freely, allowing for further observation and analysis.

Here, we fabricated FETs using 12 nm MoTe_2_ with 1-butyl-3-methylimidazolium tetrafluoroborate (BMIM-BF_4_) as a gate dielectric material. The utilization of ILG proves to be a more effective method for adjusting the channel resistance compared to the back gate with a 300 nm thick SiO_2_ layer. Despite the significant increase in the gate leakage current caused by the presence of an oxygen bond, ILG does not impact the resistance of the MoTe_2_/h-BN interface channel. This suggests that the ILG technique is capable of selectively modifying the channel resistance without affecting the interface properties between MoTe_2_ and h-BN. In our study, we observed unipolar behavior in the MoTe_2_/h-BN-BMIM-BF_4_ FET system. This means that the system predominantly conducted either positive or negative charges but not both simultaneously. The unipolar behavior suggests that the system is suitable for specific electronic applications that require controlled charge transport. These findings highlight the potential of the MoTe_2_/h-BN-BMIM-BF_4_ FET system in developing advanced electronic devices.

## 2. Materials and Experimental Methods

### 2.1. Materials

MoTe_2_ and h-BN flakes obtained in the bulk form from Sigma-Aldrich (Yongin, South Korea), were mechanically exfoliated using the standard Scotch tape method in collaboration with the HQ GRAPHENE Company (Groningen, The Netherlands) [46]. The BMIM-BF_4_ ionic liquid was also sourced from Sigma-Aldrich.

### 2.2. Device Fabrication and Measurements

We fabricated the multilayer flakes of MoTe_2_ and h-BN, using the mechanical exfoliation method with standard Scotch tape, from natural bulk crystals. Adhesive tape residues were removed by immersing the flakes in an acetone solution. During the chemical etching of SiO_2_, few-layered MoTe_2_ was coated with poly(methyl methacrylate) (PMMA) as a supporting material. The SiO_2_ layer was etched using a diluted HF solution. Subsequently, the MoTe_2_ with PMMA was separated from the SiO_2_ substrate and transferred onto a preprepared template comprising highly doped p-type SiO_2_ and h-BN. We present a circuit illustration of the ILG MoTe_2_/h-BN FET device’s structure, as well as the device under atomic force microscopy (Figure 1). Schematic diagrams of the fabricated device can be seen in Figure 1a,b depicting a line that represents a step of ~12 nm; the curve was performed using AFM. The source and drain contacts were assembled using photolithography, e-beam lithography, and etching techniques. Metal contacts were created by depositing a 30/50 nm thick Cr/Au and a 5/20 nm thick Ti/Au (top-gate) film through thermal evaporation, employing standard photolithography and e-beam lithography. The device’s fabrication process was conducted at room temperature under a vacuum of 1 × 10^−6^ Torr. A solution was prepared by mixing the ionic liquid BMIM-BF_4_ with deionized water. The MoTe_2_/h-BN device was immersed in the BMIM-BF_4_ solution at room temperature for a duration of up to one hour. Following fabrication, annealing was conducted in a vacuum (∼2 × 10^−6^ Pa) furnace tube at a temperature of 150 °C for a duration of 10 h to remove water [47,48].

An optical microscope was used to characterize the structural morphology and three-dimensional schematics of the transistor, including the thickness of the MoTe_2_ and h-BN thin films (Figure 2). Multiple devices could not undergo I–V tests because of alignment issues during fabrication. Testing of the devices was conducted without the ionic liquid drop, utilizing a probe station (6-Probe Vacuum System). The electrical measurements were performed within a closed cycle refrigerator system’s (CCRS) chamber, and the temperature was varied from room temperature to 80 K by applying a top and back gate (Figure 3). The gate voltage was applied using a double-channel source measure unit (SMU 2636). For the AC photocurrent measurements, a low-noise current preamplifier (SR 570) and a lock-in amplifier (SR 830) were employed. We further investigated the impact of the temperature and gate voltage sweep rates on the observed hysteresis during the transconductance measurements in both the forward and reverse gate sweeps. These results highlight the significant potential of ILG for advanced field-effect control in the MoTe_2_ nanochannel system, which makes it an excellent candidate for exploring nanodevice physics.

## 3. Results and Discussion

Figure 3 shows the transfer curves of the MoTe_2_/h-BN device obtained at different ambient temperatures on a SiO_2_ substrate with Cr/Au and Ti/Au metal contacts before the introduction of the ionic liquid. In particular, Figure 3a,c display the transfer characteristics of the MoTe_2_/h-BN FET with the SiO_2_ back gate and Ti/Au top gate. At a back-gate voltage of V_gs_ = −40 V, we achieved a source–drain current (*I_ds_*) of ~100 nA. This increase in *I_ds_* is attributed to a reduction in the Schottky barrier. The I_on_/I_off_ ratio was enhanced to 10^−7~(−8)^ compared to the doped sample. The pristine sample was biased by applying a voltage range of +8 to −8 V to the substrate, which served as a top gate for the device. The value of the carrier density resulted in the same observations. Figure 3b presents the output characteristics of the device at a source–drain bias of −1 V to +1 V under different temperatures of the MoTe_2_/h-BN device. Generally, the source–drain current decreases as the temperature increases from 80 to 300 K, with a slight increase around 200 K. All transfer curves exhibited a unipolar n-type semiconductor behavior. Above 300 K, the ion migration became significant, leading to pronounced hysteresis and a weak field effect. Conventional DC measurement techniques involve trapping and de-trapping processes at the MoTe_2_/h-BN/SiO_2_ interface, making it challenging to entirely eliminate hysteresis in the transfer characteristics, particularly during extended measurement periods. The mobility versus temperature is depicted at 80 K, the field-effect electron mobility is approximately 3 cm^2^ V^(−1)^ s^(−1)^, surpassing that of FETs with Cr/Au electrodes. The field-effect electron mobility gradually decreased with the temperature increasing from 80 to 300 K, followed by a sudden increase around 182–200 K, attributed to the hexagonal phase transition [43,49]. Subsequently, the field-effect electron mobility decreased with further temperature increases due to carrier-phonon scattering and possible carrier scattering from hexagonal inclusions within the phase below 200 K [50]. Notably, the impact of the charged impurities was predominantly pronounced at extremely low temperatures. In the low-temperature range, phonon scattering becomes the dominant factor, and it is anticipated that mobility will exhibit a temperature-dependent decrease, conforming to a power law relationship of μeff ∼T-α, with α being a measurable positive exponent.

The few-layered h-BN served as the topmost layer, acting as an atomically uniform potential barrier, facilitating electron coupling through the tunneling process. To fulfill this role effectively, the h-BN layer must be free from contaminants and wrinkles. Recent shot noise measurements in the metal–hBN semiconductor tunnel junctions have provided evidence that h-BN behaves as an ideal tunnel barrier [51]. Our study revealed that incorporating tunnel barriers using up to four-layer h-BN can effectively suppress the Schottky barrier. This allows for the precise and uniform adjustment of the chemical potential of the MoTe_2_ layer across the entire channel, enabling accurate electrostatic control of the Fermi level of the 2D layer. An ambipolar field effect was achieved under a finite source–drain bias, enabling fully reversible *pn* to *np* diodes through back and top gating. Electric transport measurements were conducted over a wide range of source–drain voltage bias values, while the voltage applied to the liquid gate varied around zero bias, within a few volts. Additionally, we examined the back gate in a separate configuration. The study demonstrated MoTe_2_-based FET top gating, and the liquid gate operation showed superior performance compared to standard back-gating.

The interaction between the pristine ionic-liquid BMIMBF_4_ and the channel, as well as the Ti/Au top gate, resulted in a rectification behavior characteristic of the p-junction diodes, owing to the p-type and n-type properties of MoTe_2_. We observed an enhancement in the design, particularly the increased conductive mobility limited by the top gate, which can be attributed to the reduction of the Schottky barrier at the source and drain electrodes. The relationship between the accumulation charge carrier density and applied gate voltage, also known as the capacitance of the ILG TMD transistors, remains poorly understood, and the the consistent observation of capacitance per unit area in devices based on different TMDs raises questions [49]. Moreover, the electrical breakdown of the MoTe_2_/h-BN FET was observed at a current density of 5 × 10^7^ A/cm^2^, which exceeds the current-carrying capacity of Ti by 60 times. Notably, experimental demonstrations of high-transconductance FETs based on a few layers of MoTe_2_ with full-channel ILG have been conducted. Figure 4a illustrates that at a low bias of V_ds_ = 2 V, the drain–source conductance approached zero, indicating channel current saturation and offering high transconductance (with a maximum value of *g_m_* = 45 µS/µm for V_ds_ = 2 V).

Furthermore, the ionic liquid BMIM-BF_4_ encapsulation treatment resulted in a significant shift of the threshold voltages, but the necessary conditions for low-power integrated circuits have not yet been demonstrated in this specific device geometry. The substantial shift towards a higher threshold voltage indicates the effectiveness of the BMIM-BF_4_ dopant at converting the material to p-type. The MoTe_2_/h-BN FET showed ambipolar behavior with a high on/off ratio > 103 at 200 K, along with large carrier mobility (electron mobility of 345 cm^2^ V^−1^ s^−1^ and hole mobility of 285 cm^2^ V^−1^ s^−1^) at 80 K. Figure 4b illustrates that these results were achieved using a low-resistance ionic liquid gate with h-BN encapsulation and Ti/Au contacts on the MoTe_2_. To observe intrinsic, carrier scattering-limited conduction in both the electron and hole channels, a significant reduction of the Schottky barriers between the channel and Ti contact electrodes through ILG is essential. However, the implementation of ILG in atomic 2D TMDC devices still requires considerable effort because of challenges related to integration, reliability, and low operating speed [52].

We specifically selected the excitation wavelength to be 532 nm, which provides a large Raman intensity on the monolayers, and we maintained the low laser power of 1 mW/μm^2^ to avoid inducing the 2H phase or damaging the sample through laser irradiation. When biasing the top gate to 0 V, the initial exfoliated 2H MoTe_2_ was observed. In Figure 4c, the A_1g_ peak of the pristine MoTe_2_/h-BN was detected at 168.5 cm^−1^. Following the treatment with BMIM-BF_4_, the A_1g_ peak exhibited a shift of 1.6 cm^−1^ towards a lower wavenumber, while the linewidth remained unchanged. The peaks demonstrate a simultaneous intensity drop and linewidth broadening when the gate bias exceeded 2 V—an indication of ongoing bond breaking and the disappearance of the 2H phase. Furthermore, the Raman spectra reveal a new rising peak at 168.5 cm^−1^, which we confirm corresponds to the characteristic A_g_ mode of the 2H phase. This evidence conclusively points to the formation of the 2H phase under the top-gate control. These significant findings provide confirmation that the use of BMIM-BF_4_ ILG outperforms the conventional FET treatment in enhancing the transconductance properties and stability of MoTe_2_/h-BN.

## 4. Conclusions

In summary, we presented a facile and effective method for high-performance 2D insulators for MoTe_2_ FETs using BMIM-BF_4_ ILG encapsulation. We focused on the exploration of novel approaches and improvements over the control of the electronic properties of these devices through the use of ionic liquids, specifically to achieve unipolar behavior in MoTe_2_/h-BN FETs.

The resulting few-layered MoTe_2_/h-BN FETs exhibited unipolar p-type behavior with excellent electrical transconductance properties, a high on/off ratio, and significant enhancement in hole mobility. Moreover, the incorporation of doped samples imparted exceptional electrical and optical durability. Our approach to encapsulation and chemical treatment offers a practical p-doping method for functionalizing 2D materials with environmental stability, making them promising candidates for post-silicon electronic and optoelectronic device applications.

## Figures and Tables

**Figure 1 nanomaterials-13-02559-f001:**
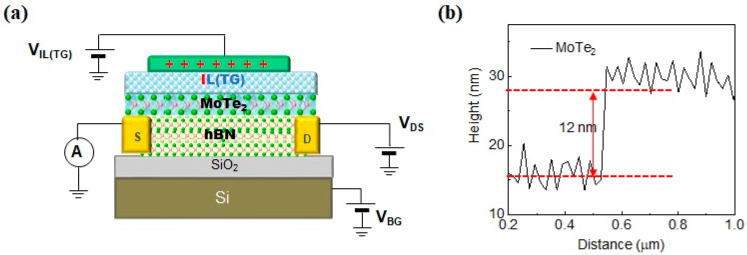
(**a**) Circuit illustration of the structure of a h-BN/MoTe_2_ ILG FET with Ti/Au contacts; (**b**) the AFM line scan indicates a flake thickness of ~12 nm, corresponding to 8~10 layers.

**Figure 2 nanomaterials-13-02559-f002:**
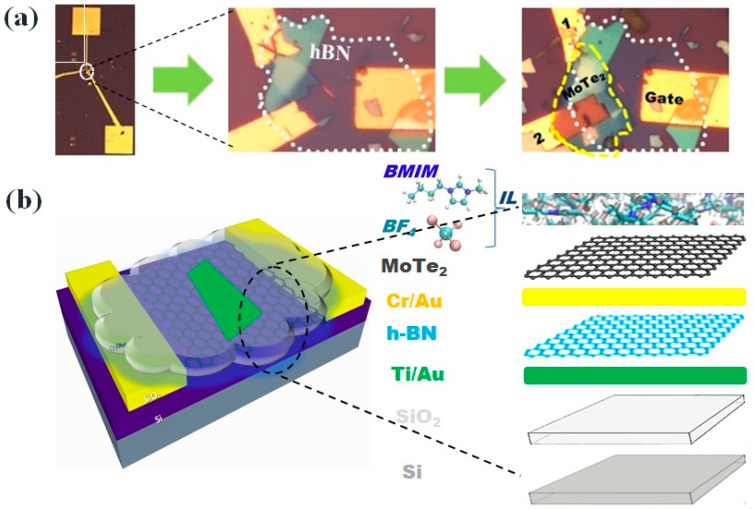
(**a**) Optical images of the MoTe_2_/h-BN ILG FETs; (**b**) three-dimensional schematic of the transistor with an ionic liquid logics device.

**Figure 3 nanomaterials-13-02559-f003:**
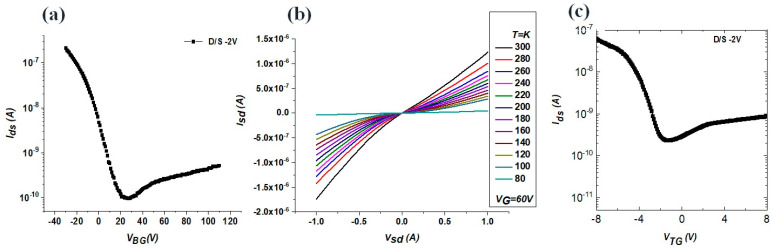
Transfer characteristic of the MoTe_2_/h-BN FET before dropping the ionic liquid (**a**) at back gate and (**b**) measured within three days after loading the sample in the vacuum setup at different temperatures; (**c**) transport measurements recorded at different a gate (top gate, before droping the IL).

**Figure 4 nanomaterials-13-02559-f004:**
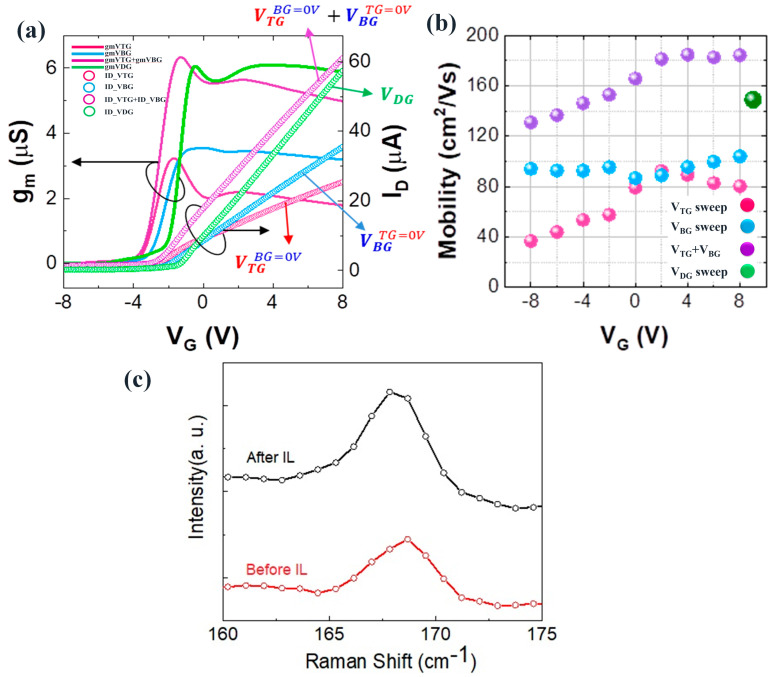
(**a**) The transconductance was determined from the I_ds_-V_ds_ characteristics at a fixed V_ds_ of 2 V; (**b**) the device exhibited low-field mobility and field-effect mobility, which confirms the performance enhancement achieved through the interfacial h-BN layer and ionic liquid gate passivation; (**c**) Raman spectrum of the MoTe_2_/h-BN heterostcture (black line represents after and the red line before ionic liquid gating).

## Data Availability

All the data are included in the main text.

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
