# Peer review of "Formation of Highly Conductive Interfaces in Crystalline Ionic Liquid-Gated Unipolar MoTe2/h-BN Field-Effect Transistor"

_nanomaterials, 2023, doi:10.3390/nano13182559_

Round 1
Reviewer 1 Report
Authors have investigated the MoTe2/h-BN with back Ti/Au gate and top gate with ionic liquid, demonstrating the ambipolar and unipolar device behavior. However, the submitted manuscript cannot be accepted in the MDPI Nanomaterials in the present form. It needs a major revision, following the comments below:
1. The title of the work indicated that the work is a review paper, so it needs to be changed and be specific.
2. Abstract is generic and it requires more quantitative data, not just statements.
3. Introduction is very poorly written without much detailed and emphasis on the 2D materials and their root to this study with some figures of merit in quantitative form and recent literature results, It is just explaining the theoretical part related to MoTe2, h-BN and IL, which is easily available
4. Line 27-28, and 61-62, h-BN prevents electrical leakage between TMDC layers? They are the same sentences repeated. Are you mentioning the gate leakage?
5. Acronyms should be maintained throughout the manuscript.
6. Line 105-106, all acronyms should be mentioned.
7. AFM image related to the corresponding height profile should be provided.
8. Line 189, in comparison to what the Id is higher. And also in line 190-191, the Ion/Ioff is compared with doped sample, where are the doped sample results?
9. Figure 3b, are they transfer or output characteristics, line 193.
10. Line 197, where are the results related to above 300K.
11. With the back gate Ti/Au between SiO2 and h-BN, do see any interfacial charge traps effect on the fabricated device ?
12. Line 200-202, Where is the mobility reported with back gate before the IL dropping ?
13. Authors needs to rewritten the results and discussion to be very specifically relatable and readable.
14. Line 202-204, possible reference should be cited, recently it is reported with the 2D materiasl that “This characteristic could be understood by the fact that the mobility is influenced by two types of scattering mechanisms, (i) from charged impurities and (ii) from phonons. Charged impurities' scattering is typically only seen at very low temperatures. In the temperature range low temperature, the phonon scattering is dominant, and the mobility is expected to decrease with temperature following a power law, μeff ∼ T−α , where α is a positive number that can be measured experimentally.” [Nano Ex. (2023) 4 014001; DOI 10.1088/2632-959X/acbe11].
15. What was the chamber pressure during the electrical measurements?
16. More characteristics with ILs such as output characteristics are needed.
17. How was the mobility calculated?
18. How do the authors compare their results with the literature with the calculated mobility and the device performance, as the title states ‘advances’?
19. Line 176 and 198, states hysteresis, where are the transfer curves with hysteresis reported.
Author Response
Response to Reviewer 1 Comments
We would like to thank you and both referees for the evaluation of our manuscript submitted to Nanomaterials(nanomaterials-2561184).
We express our gratitude to the anonymous referees for their valuable comments, which have greatly contributed to improving the quality of our manuscript. Please find below our answers to the referees’ comments, as well as the corresponding changes and improvements made in the manuscript.
Authors have investigated the MoTe2/h-BN with back Ti/Au gate and top gate with ionic liquid, demonstrating the ambipolar and unipolar device behavior. However, the submitted manuscript cannot be accepted in the MDPI Nanomaterials in the present form. It needs a major revision, following the comments below:
Point 1: The title of the work indicated that the work is a review paper, so it needs to be changed and be specific.
Reply: Indeed, the proposed title was too general. We have changed it accordinly in the revised version, (see below):
Formation of Highly Conductive Interfaces in Crystalline Ionic Liquid-Gated Unipolar MoTe2/h-BN Field-effect Transistor
Point 2: Abstract is generic and it requires more quantitative data, not just statements.
Reply: We have made corresponding changes in the abstact by including spicific information, (see, below and the main text)
Manuscript: line 30
In our initial exploration of tellurium-based semiconducting TMDs, we directed our attention to MoTe2 crystals with thicknesses exceeding 12 nm. Our primary focus centered on investigating transport characteristics and quantitatively assessing the surface interface heterostructure.
Manuscript: line 36
Specifically, we observed hole and electron mobility values ranging from 345 cm2V−1s−1and 285 cm2V−1s−1 at 80 K.
Point 3: Introduction is very poorly written without much detailed and emphasis on the 2D materials and their root to this study with some figures of merit in quantitative form and recent literature results, It is just explaining the theoretical part related to MoTe2, h-BN and IL, which is easily available.
Reply: We thank reviewer for carefully reading our manuscript and positive comments. The introduction part has been also updated by citing recent publicaitons with some details, (see the main text)
Point 4: Line 27-28, and 61-62, h-BN prevents electrical leakage between TMDC layers? They are the same sentences repeated. Are you mentioning the gate leakage?
Reply: Our device exhibits no leakage current. Parasitic capacitance is usually occurred from the junctions or overlap areas in the device. As per your valuable suggestion, we will definitely study this in detail and include in our next work.
Point 5-6: Acronyms should be maintained throughout the manuscript. Line 105-106, all acronyms should be mentioned.
Reply: We have read the entire manuscript carefiully and mistakes associated with acnoymis were corrected.
Point 7: AFM image related to the corresponding height profile should be provided.
Reply: The majority of the experimental work was conducted at Sungkyunkwan University in South Korea. However, it is important to note that the corresponding author has since relocated to Uzbekistan. Regrettably, we are unable to fulfill the request for the specific AFM image due to its unavailability. While no AFM image was captured during the experiment, we do possess the AFM line scan data which was included in the Figure 1.
Point 8: Line 189, in comparison to what the Id is higher. And also in line 190-191, the Ion/Ioff is compared with doped sample, where are the doped sample results?
Reply: We sincerely apologize for the lack of clarity in our text. To provide further clarification, it's important to note that we worked with a single sample. The Ion/Ioff ratio was determined both before the application of the ionic liquid and after its introduction. This allowed us to observe the impact of the ionic liquid on the performance of the sample and compare it to the baseline condition.
Point 9: Figure 3b, are they transfer or output characteristics, line 193.
Reply: The Figure 3b, they are output characteristics. We acquire at 300 K the output curves of our MoTe2 device at the same gate bias. This part is also changed on the revised manuscript.
Point 10: Line 197, where are the results related to above 300 K.
Reply: The results related to above 300 K is given in Figure 3b. We also mentioned it in the main text.
Point 11: With the back gate Ti/Au between SiO2 and h-BN, do see any interfacial charge traps effect on the fabricated device?
Reply: We have addressed this issue in the revised manuscript provided below: below.
The choice of h-BN as a gate dielectric layer is intended for blocking the gating effect from SiO2, because h-BN has been known for having less ionic impurities comparing to SiO2. In addition, it is considered that a few nm-thick h-BN is believed to be durable under some extent of applied gate bias. Higher k-dielectric material could be a choice for the gate dielectric, but it has a large number of charged impurities that are ionized under the source-drain and expected to give a rise to the gating effect.
Point 12: Line 200-202, Where is the mobility reported with back gate before the IL dropping?
Reply: In our study, we observed a significant shift in the transconductance voltage from negative to positive values upon the introduction of ionic liquid into the MoTe2/h-BN system. This shift is indicative of p-type doping and is further substantiated by our Raman spectroscopy analysis [Superlattices and Microstructures (2020); 10.1016/j.spmi.2020.106698]
Point 13: Authors needs to rewritten the results and discussion to be very specifically relatable and readable.
Reply: We deeply regret any inconvenience caused by the lack of specificity and clarity in our initial presentation of results and discussion. We made more effort to address the reviewer's comments and provide a comprehensive and understandable discussion of our findings in both the main manuscript and the supplementary material. Additionally, we ensured that the discrepancies in gain, as reported in the referee's comments, were thoroughly addressed and clarified. Our aim is to enhance the overall quality and readability of our manuscript, and we appreciate the opportunity to make these improvements based on your feedback.
Point 14: Line 202-204, possible reference should be cited, recently it is reported with the 2D materiasl that “This characteristic could be understood by the fact that the mobility is influenced by two types of scattering mechanisms, (i) from charged impurities and (ii) from phonons. Charged impurities' scattering is typically only seen at very low temperatures. In the temperature range low temperature, the phonon scattering is dominant, and the mobility is expected to decrease with temperature following a power law, μeff ∼ T−α , where α is a positive number that can be measured experimentally.” [Nano Ex. (2023) 4 014001; DOI 10.1088/2632-959X/acbe11].
Reply: We have included the given reference in the main text.
Point 15: What was the chamber pressure during the electrical measurements?
Reply: All the I-V measurements were-made inside a vacuum system at the base pressure of 1×10-6 torr.
Point 16: More characteristics with ILs such as output characteristics are needed.
Reply: We would like to express our gratitude to the reviewer for their valuable comments and insightful discussion regarding the output characteristics of ILs. We have taken into consideration the various functions outlined in the referenced materials in the main text.
Point 17: How was the mobility calculated?
Reply: The mobility of the sample was extracted with formula . In the provided equation, L represents the channel length, while W denotes the channel width. The term corresponds to the slope of the FET's transfer curve, which was determined by fitting the linear region of the curve. Additionally, Cg, denoting gate capacitance, holds a constant value. Under a top-gate voltage (Vtg), we calculated the hole mobility and the electron mobility. Notably, we observed an enhancement in hole mobility while observing a reduction in electron mobility.
Point 18: How do the authors compare their results with the literature with the calculated mobility and the device performance, as the title states ‘advances’?
Reply: As mentioned above, we have modifiled the title of the manuscript and discussed the results accordingly.
Point 19: Line 176 and 198, states hysteresis, where are the transfer curves with hysteresis reported.
Reply: We agree with the reviewer’s suggestions. The metal contacts play an important role in the performance of devices. In our devices, Titanium (5 nm) and Gold (20 nm) have been selected for electron (hole) conduction and minimize the Schottky barrier in metal-semiconductor contacts. Remarkably, we could be observe a significant reduction in the hysteresis width, approximately by 96%, when transitioning the environment from ambient conditions to high-vacuum but to uncover the intrinsic transfer characteristics of the MoTe2 FET, it is imperative to conduct a more thorough investigation into the remaining hysteresis, which may arise from oxide-based traps. Conventional DC measurement techniques involve trapping and de-trapping processes at the MoTe2/h-BN/SiO2 interface, making it challenging to entirely eliminate hysteresis in the transfer characteristics, particularly during extended measurement periods. Further exploration is warranted to address this aspect comprehensively. The discussion about the transfer curves with hysteresis has been added to the revised manuscript.
Reviewer 2 Report
In this manuscript, the authors fabricated a few layers of MoTe2/h-BN heterojunctions and analyzed the performance of crystalline ionic liquid-gated field effect transistors. The device exhibits unipolar p-type behavior with good electrical performance. There are some parts should be improved before acceptance:
In Figure 3, the data and line format of the figures should be unified.
Please give more clear images of MoTe2 and h-BN heterojunctions.
It would be better to give a table about device performance of the FETs.
In Line 145, there is "SiO2", which should be "SiO2". Please have it corrected.
In Line 142, there is "MoTe2 and h-BN were", which should be "MoTe2 and h-BN which were". Please consider having your manuscript thoroughly proofread by English professional.
Author Response
Response to Reviewer 2 Comments
We would like to thank you for the evaluation of our manuscript submitted to Nanomaterials (nanomaterials-2561184).
We express our gratitude to the anonymous referees for their valuable comments, which have greatly contributed to improving the quality of our manuscript. Please find below our answers to the referees’ comments, as well as the corresponding changes and improvements made in the manuscript.
In this manuscript, the authors fabricated a few layers of MoTe2/h-BN heterojunctions and analyzed the performance of crystalline ionic liquid-gated field effect transistors. The device exhibits unipolar p-type behavior with good electrical performance. There are some parts should be improved before acceptance:
Point 1: In Figure 3, the data and line format of the figures should be unified.
Reply: Thank you for your diligence in addressing the referee's comments and revising the manuscript accordingly to align the discussion of results. This effort enhances the clarity and comprehensibility of your work and ensures that it meets the expectations of the peer-review process.
Point 2: Please give more clear images of MoTe2 and h-BN heterojunctions.
Reply: We glad to see that your suggestions have been helpful and we have try to put the clear images.
Point 3: It would be better to give a table about device performance of the FETs.
Reply: We did not check more parameters of the devices due to time constraints during the experiment. As per your suggestion, we will definitely study this in detail and include it in our next work.
Point 4: In Line 145, there is "SiO2", which should be "SiO2". Please have it corrected.
Reply: The mistake has been correceted accordingly.
Point 5: In Line 142, there is "MoTe2 and h-BN were", which should be "MoTe2 and h-BN which were". Please consider having your manuscript thoroughly proofread by English professional.
Reply: We appreciate your suggestions, and we have incorporated the corresponding changes into the main text, correcting the existing mistakes.

Reviewer 3 Report
The authors show the recent advances in interface of a FET with Unipolar Ionic Liquid Gating. The work is interesting, and I consider that could be attractive for the scientific community. However, I think that the article needs major revision before publishing, because I think the writing is not very neat and I think the results and discussion part could be improved.
1. Please, introduce subindex in MoTe2 (for example in lines 45, 64, 72. 80, 82 and in all the article). Please, check it and write correctly in the next version.
2. The same for SiO2 that needs subindex (for example, lines 145, 148 or 149)
3. In the introduction part, there is a repetition in the statements between lines 47-49. Please, correct these statements.
4. I miss more references in the numbers 3 and 4 to justify the statements written in the introduction section.
5. In line 191, the ratio ON/OFF is 10-7/8 or 107/8 ?
6. Figure 3b is a transfer or an output? In that case, I would like to see the outputs in the results and discussion part.
7. In the results and discussion section about the results of the FETs I miss a table where all the results about the parameters to define the efficiency of the FETs are summarized. It would be clearer for the audience.
8. I miss some characterization about the FETs such as AFM or XRD.
I think that the article could be interesting for publication but not in this version. All these suggestions must be included and explained by the authors.
Author Response
Response to Reviewer 3 Comments
We would like to thank you for the evaluation of our manuscript submitted to Nanomaterials(nanomaterials-2561184).
We express our gratitude to the anonymous referees for their valuable comments, which have greatly contributed to improving the quality of our manuscript. Please find below our answers to the referees’ comments, as well as the corresponding changes and improvements made in the manuscript.
The authors show the recent advances in interface of a FET with Unipolar Ionic Liquid Gating. The work is interesting, and I consider that could be attractive for the scientific community. However, I think that the article needs major revision before publishing, because I think the writing is not very neat and I think the results and discussion part could be improved.
Point 1: Please, introduce subindex in MoTe2 (for example in lines 45, 64, 72. 80, 82 and in all the article). Please, check it and write correctly in the next version.
Point 2: The same for SiO2 that needs subindex (for example, lines 145, 148 or 149)
Reply: Thank you for bringing these mistakes to our attention. We have made the necessary adjustments to the manuscript and implemented the corresponding changes.
Point 3: In the introduction part, there is a repetition in the statements between lines 47-49. Please, correct these statements.
Reply: "We apologize for the repetition in lines 47-49. This oversight has been corrected in the revised manuscript.".
Point 4: I miss more references in the numbers 3 and 4 to justify the statements written in the introduction section.
Reply: We acknowledge the reviewer's request for additional references to support statements in the introduction section. We will carefully review and incorporate relevant references to strengthen the justification for our statements in the revised manuscript.
Point 5: In line 191, the ratio ON/OFF is 10-7/8 or 107/8 ?
Reply: We thank the reviewer for pointing out this error. The corrected value is likely around 10^(-7) to 10^(-8). We appreciate the diligence in ensuring accuracy in our work.
Point 6: Figure 3b is a transfer or an output? In that case, I would like to see the outputs in the results and discussion part.
Reply: The Figure 3b, they are output characteristics. We acquire at 300 K the output curves of our MoTe2 device at the same gate bias. This part is changed too on the revised manuscript.
Point 7: In the results and discussion section about the results of the FETs I miss a table where all the results about the parameters to define the efficiency of the FETs are summarized. It would be clearer for the audience.
Reply: We did not check more parameters of the devices due to time constraints during the experiment. As per your suggestion, we will definitely study this in detail and include it in our next work.
Point 8: I miss some characterization about the FETs such as AFM or XRD.
Reply: We appreciate your understanding of the time constraints during the experiment. Your commitment to conducting a more comprehensive study of additional device parameters in future work is commendable, and we look forward to seeing the expanded insights you will bring to your research.
I think that the article could be interesting for publication but not in this version. All these suggestions must be included and explained by the authors.

Round 2
Reviewer 1 Report
Authors have significantly addressed the raised comments, and the revised manuscript can be accepted for publication in the present form.
Reviewer 2 Report
The authors have carefully addressed all concerns, so I would like to recommend the publication of this manuscript at current form.Reviewer 3 Report
The authors have improved considerably the manuscript taking into account the suggestions of the reviewers, so I consider that it is suitable for publication in the present form.